# Modelling the transmission dynamics of H9N2 avian influenza viruses in a live bird market

Francesco Pinotti [1] ✉, Lisa Kohnle [2], José Lourenço[3], Sunetra Gupta [1], Md. Ahasanul Hoque[4], Rashed Mahmud[4], Paritosh Biswas[4], Dirk Pfeiffer [2,5] & Guillaume Fournié[5,6,7]

H9N2 avian influenza viruses (AIVs) are a major concern for the poultry sector and human health in countries where this subtype is endemic. By fitting a model simulating H9N2 AIV transmission to data from a field experiment, we characterise the epidemiology of the virus in a live bird market in Bangladesh. Many supplied birds arrive already exposed to H9N2 AIVs, resulting in many broiler chickens entering the market as infected, and many indigenous back-yard chickens entering with pre-existing immunity. Most susceptible chickens become infected within one day spent at the market, owing to high levels of viral transmission within market and short latent periods, as brief as 5.3 hours. Although H9N2 AIV transmission can be substantially reduced under moderate levels of cleaning and disinfection, effective risk mitigation also requires a range of additional interventions targeting markets and other nodes along the poultry production and distribution network.

H9N2 Avian influenza virus (AIV) is considered to be the most prevalent AIV in poultry globally[1]. Despite being classified as a low pathogenic virus, H9N2 AIV is responsible for substantial economic loss for the poultry industry[2,3]. Infection is typically associated with moderate to severe respiratory symptoms, delayed growth, reduced egg production and increased mortality, especially when co-infection with other pathogens is involved[4]. Some H9N2 AIV lineages are known to be zoonotic, with resulting symptoms being typically mild. Co-circulation with other AIV subtypes may lead to the emergence of reassortant viruses with increased pathogenicity and/or zoonotic potential[5–7]. H9N2 appears to be involved in the origin of several novel zoonotic AIVs, whose number has been rapidly increasing since 2013[8]. AIVs with H9N2-derived genes include H7N9[9], H5N1, H10N8[10–12] and, more recently, H3N8[13].

In many Asian countries, the prevalence of H9N2 AIVs is particularly high in live bird markets (LBMs), with estimates in Bangladeshi markets as high as 80%[14,15]. LBMs play a central role in the marketing of poultry in developing countries, being the place of choice for many people to purchase meat for consumption. At the same time, the high prevalence of AIV infection among traded poultry is concerning due to the risk of zoonotic spillover to humans[5,16,17]. In LBMs, the latter may be exposed to AIV through contaminated dust particles, water, surfaces and the slaughtering of infected birds. LBMs are also known to promote the mixing and evolution of AIVs, in that they enable the intermingling of multiple poultry species from many distant locations and diverse farming systems[18–20]. Over the last 25 years, public health concerns around LBMs have prompted health authorities in several Asian countries to take steps to control AIV transmission in these settings; adopted measures included enhanced hygiene protocols, bans on overnight poultry storage, as well as periodic rest days[21–26]. Temporary and permanent market shutdowns have also been employed in response to outbreaks of emerging zoonotic AIVs[27].

The central role played by LBMs in disseminating AIVs, including H9N2 viruses, calls for a better understanding of AIV transmission

[1]Department of Biology, University of Oxford, Oxford, UK. [2]City University of Hong Kong, Hong Kong SAR, Hong Kong. [3]CBR (Biomedical Research Centre), Universidade Católica Portuguesa, Oeiras, Portugal. [4]Chattogram Veterinary and Animal Sciences University, Chittagong, Bangladesh. [5]Royal Veterinary College, London, UK. [6]INRAE, VetAgro Sup, UMR EPIA, Université de Lyon, Marcy l'Etoile 69280, France. [7]INRAE, VetAgro Sup, UMR EPIA, Université Clermont Auvergne, Saint Genès Champanelle 63122, France. ✉e-mail: francesco.pinotti@biology.ox.ac.uk

dynamics in these settings, which is paramount to design and implement effective and appropriate interventions. Previous field research focused on specific epidemiological aspects of AIV transmission, e.g. contamination in the environment[28–31], or involved cross-sectional investigations of AIV circulation in LBMs[15]. Unfortunately, linking results from these studies to viral dynamics is not straightforward. Challenge and transmission experiments in which live virus is inoculated artificially into chickens, and eventually transmitted onwards[15,32], allow us to estimate important properties of AIV epidemiology. However, because these experiments are conducted within a controlled environment, it remains difficult to draw general conclusions about AIV transmission in LBMs.

Here we aimed to fill these gaps by modelling H9N2 AIV transmission in an LBM. Mathematical modelling has proven useful to study AIV transmission dynamics in LBMs, but such investigations have been mostly theoretical so far[22]. Our work is instead grounded on a longitudinal dataset of H9N2 AIV acquisition in exotic and indigenous chickens in an LBM in Chattogram, Bangladesh[33]. Using Bayesian methods, we estimated quantities of epidemiological relevance, including H9N2 AIV transmission rate, host-specific latent periods, and quantified within-market prevalence as well as the likelihood of prior chicken exposure to H9N2 before entering the LBM. Finally, we leveraged these results to assess the impact of a range of hypothetical veterinary public health interventions on H9N2 AIV transmission.

## Results

### Parameter inference

Our model simulated the transmission of avian influenza viruses (AIVs) among chickens in an LBM in Chattogram, Bangladesh. There, a fast turnover of poultry (Supplementary Fig. 1A) drew together a steady supply of susceptible animals and unsold chickens offered for sale in previous days, thus creating opportunities for the amplification of AIVs.

Following our experimental design, explained in detail in ref. 33, we focused on exotic broiler (BR) and local, backyard-raised (BY) chicken types, which represent a large share of chickens traded daily in the LBM (Supplementary Fig. 1B). We further distinguished between chickens traded along conventional (control, c) and altered (intervention, i) marketing channels. The latter involved purchasing chickens from farms rather than from traders at the market, thus avoiding intermediate transport and storage steps. We assumed these chickens could differ in terms of prior exposure to AIVs, possibly due to our intervention, which consisted in applying strict biosecurity measures during the collection and transport of farm-acquired chickens before introducing them to the LBM. Control chickens, instead, were recruited from market vendors among those recently supplied by mobile traders.

We fitted our model to H9N2 Polymerase chain reaction (PCR) positivity data[33]. We considered samples with a cycle threshold ($Ct$) <40 as positive, in accordance with the laboratory protocols of the Australian Animal Health Laboratory (Geelong, Australia, http://www.csiro.au/places/AAHL). A more conservative criterion for positivity ($Ct < 33$) was also considered throughout the analysis. We obtained posterior estimates and credible intervals (C.I.) for thirteen parameters listed in Table 1; these include H9N2 AIV transmissibility $\beta$, latent periods $T_{E,b}$ for types $b = $ BR and BY (panels Fig. 1A–C, respectively) and probabilities of prior exposure $\rho_{g,b}$ for different combinations of chicken type and recruitment group $g = c, i$. A description of prior distributions for each fitted parameter can be found in Supplementary Table 1, while posterior marginal distributions and pairwise plots are shown in Supplementary Fig. 2. Goodness of fit was checked through posterior predictive checks (Supplementary Fig. 3).

From our model's output, we found a shorter latent period in exotic broiler compared to backyard chickens (Fig. 1B, C), lasting an

**Table 1 | Fitted parameters**

| Name | Description |
|---|---|
| $\beta$ | Transmissibility |
| $\sigma_{BR}$ | Latent to infectiousness rate (broiler) |
| $\sigma_{BY}$ | Latent to infectiousness rate (backyard) |
| $\mu$ | Recovery rate |
| $\eta$ | Positivity waning rate |
| $\lambda_{BR}$ | Inverse scale past exposure time (broiler) |
| $\lambda_{BY}$ | Inverse scale past exposure time (backyard) |
| $\kappa_{BR}$ | Shape past exposure time (broiler) |
| $\kappa_{BY}$ | Shape past exposure time (backyard) |
| $\rho_{c,BR}$ | Prior exposure prob. (control, broiler) |
| $\rho_{i,BY}$ | Prior exposure prob. (intervention, broiler) |
| $\rho_{c,BR}$ | Prior exposure prob. (control, backyard) |
| $\rho_{i,BY}$ | Prior exposure prob. (intervention, backyard) |

Description of fitted parameters.

average of 5.3 h for exotic broiler, and 1 day for backyard chickens. With a more conservative criterion for positivity ($Ct < 33$ instead of $Ct < 40$), these estimates increased to 6.1 h and 1.3 days. In these exercises we assume that infected chickens would test positive only from the point where they start shedding, i.e. since the onset of infectiousness. We also found remarkably high levels of transmission in the LBM, which translated into more than 80% of chickens entering the market as susceptible, becoming infected within 20 h, regardless of whether we set the threshold for positivity to $Ct = 40$ or $Ct = 33$ (Fig. 1D). However, we estimated higher transmission under $Ct = 40$, where more than 80% of poultry became infected within 10 h, in contrast to nearly 55% for $Ct = 33$. This was likely due to the fact that the latter threshold corresponds to less positive samples in the data with respect to $Ct = 40$.

We also obtained posterior estimates for the proportions of chickens that were already infected (i.e. latent or infectious, E+I) or immune to H9N2 (R) at recruitment, for any combination of chicken type and recruitment group (Fig. 1E, F, show exotic broilers and backyard chickens, respectively). Interestingly, we found different patterns across chicken types: in the case of exotic broilers, most chickens with prior exposure to H9N2 were either infectious or latent, with only a minor proportion of them being immune (Fig. 1E). In contrast, most previously exposed backyard chickens were immune to H9N2 (Fig. 1F). Our results thus suggest that prior infection occurs close to marketing age for broilers, whereas in backyard chickens it may occur further in the past (see Supplementary Fig. 4 for distributions of time since exposure). These findings are consistent with known rearing practices and ages at sale of each chicken type: broilers are selectively bred to grow rapidly, and are sold for meat after just 28–31 days after hatching[17]. Backyard chickens are instead raised for meat and eggs in rural households and can reach much older ages. For context, backyard chickens in our dataset were aged between 90 and 720 days.

We also found differences between control and intervention chickens already at recruitment. In the broiler case, intervention chickens were less likely to be already exposed at recruitment compared to their control counterparts (odds ratio 0.44–0.58, depending on $Ct$, see Fig. 1E). However, the reverse was the case in backyard chickens, with a larger proportion of intervention chickens being already exposed to H9N2 compared to controls (odds ratio 2.37–2.13, depending on $Ct$).

The relative importance of external introductions of infected chickens and local transmission is assessed in Supplementary Fig. 5. We find comparable proportions of LBM-acquired infections caused by

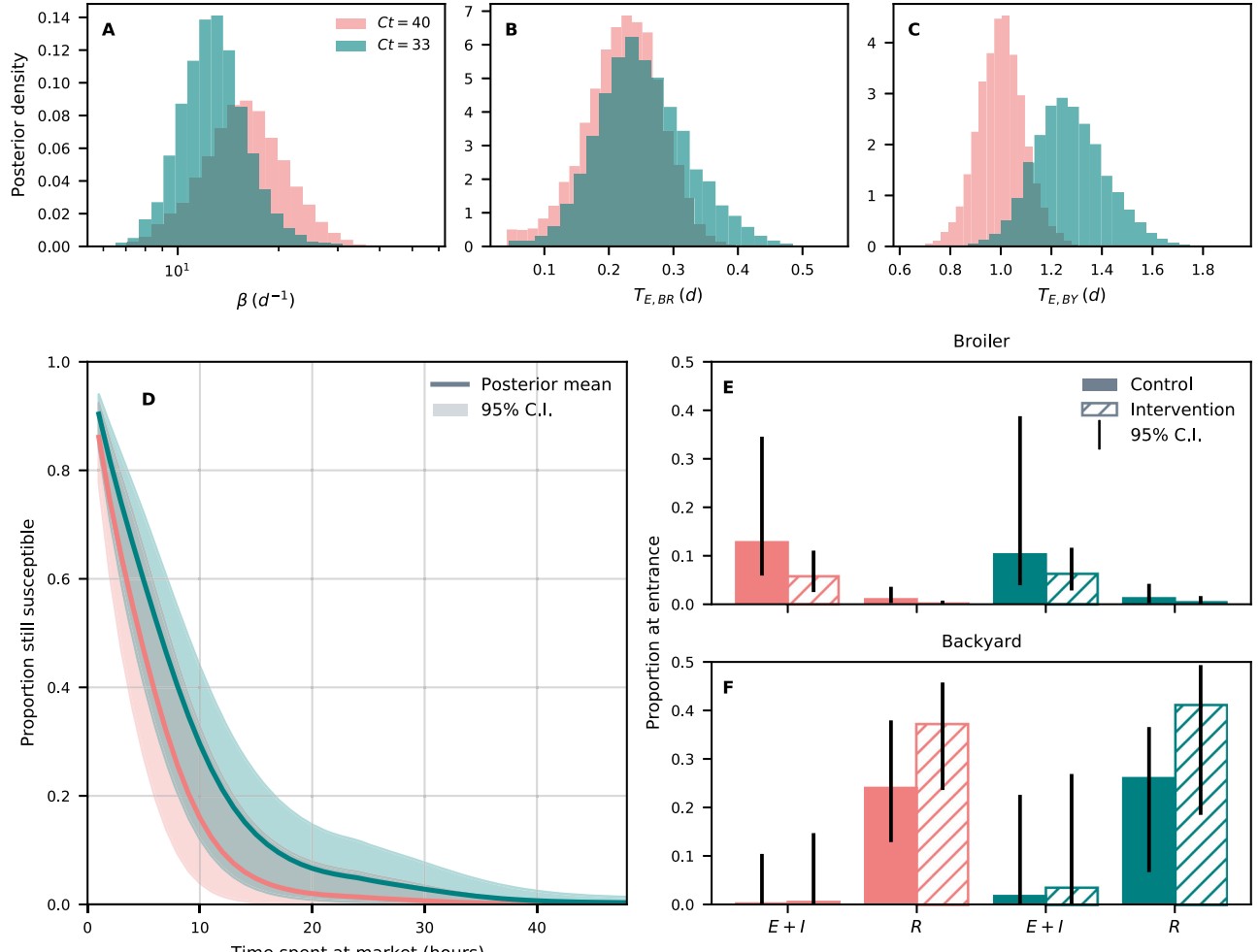

**Fig. 1 | Model fit results.** Posterior distributions for $\beta$ (**A**), $T_{E,BR}$ (**B**), and $T_{E,BY}$ (**C**) obtained from fits to $Ct = 40$ (coral) and $Ct = 33$ (teal) data. **D** Average posterior probability of a chicken remaining susceptible after a given amount of time spent at the market and 95% C.I. (shaded area). **E**, **F** Average proportions of exotic broiler and backyard chickens in either control (solid) or intervention (dashed) groups entering the market as latent or infectious ($E + I$) or recovered ($R$). 95% C.I. are denoted with black lines. For both fits we set prior hyper-parameters $l_\beta = 0.005$ and $\bar{T}_{EI} = 5$ days (see Supplementary Methods). Results in (**D**) are based on 30000 simulations based on 3000 samples from the posterior, each simulation tracking $10^6$ experimental chickens; all other panels are based on 8360 posterior samples, obtained after discarding the first 10000 MCMC iterations and keeping one sample every 1000th iteration.

chickens infected before and after entering the LBM. Moreover, we estimate that within-LBM transmission is high enough ($R_0 = 3.7–4.9$ depending on $Ct$ threshold) to ensure long-term persistence of H9N2 AIV within the LBM in the absence of further external introductions (Supplementary Fig. 6).

Our results, in particular posterior estimates of latent periods and probability of prior exposure, are robust to prior assumptions on transmissibility $\beta$ and time to viral clearance–i.e. the sum of infectious and latent periods–(Supplementary Figs. 7 and 8). Finally, our inferential procedure was able to recover model parameters in the context of synthetic data simulated from the same generative process used for inference (Supplementary Fig. 9). In particular, we show that inference succeeds in a range of scenarios where model parameters differ across chicken types and recruitment groups and in the presence of moderately biased prior assumptions about shedding time.

### Modelling interventions

In the last 20 years, LBMs have often been the target of veterinary public health interventions aiming to mitigate AIV transmission. Yet, the effectiveness of individual measures is difficult to assess and are likely to vary between different social, economic and political contexts. Here, we leveraged our inferential results to evaluate the impact of

various potential control measures to reduce H9N2 transmission in an LBM. In doing so, we considered different modes of transmission, namely direct and mediated by environmental contamination, and assessed the sensitivity of our results to each assumption. With environmentally-driven transmission, the force of infection was assumed to be proportional to environmental contamination $I_{env}(t)$; $I_{env}(t)$ accumulates due to shedding from infectious chickens and decays progressively at rate $\Theta$. We did not attempt to fit this model to data; rather, we mapped each value of "direct" transmissibility $\beta$ from previous posterior samples into an appropriate value of environmental transmissibility ($\beta_{env}$) yielding similar prevalence levels. The exact mapping, suggested by[22] and derived in the "Materials and Methods" section, is:

$$\beta \longrightarrow \beta_{env} = \beta \cdot (1 - e^{-\Theta}). \qquad (1)$$

Note that this relation depends on the decay rate $\Theta$ and that a slower decay corresponds to a smaller $\beta_{env}$, which compensates for the longer persistence in the environment. Here we consider three values of $\Theta$, namely $\Theta^{-1} = 10, 3, 1$ days, corresponding to slow, intermediate and fast decay, respectively. These values are based on actual estimates from the scientific literature and capture a broad range of environmental

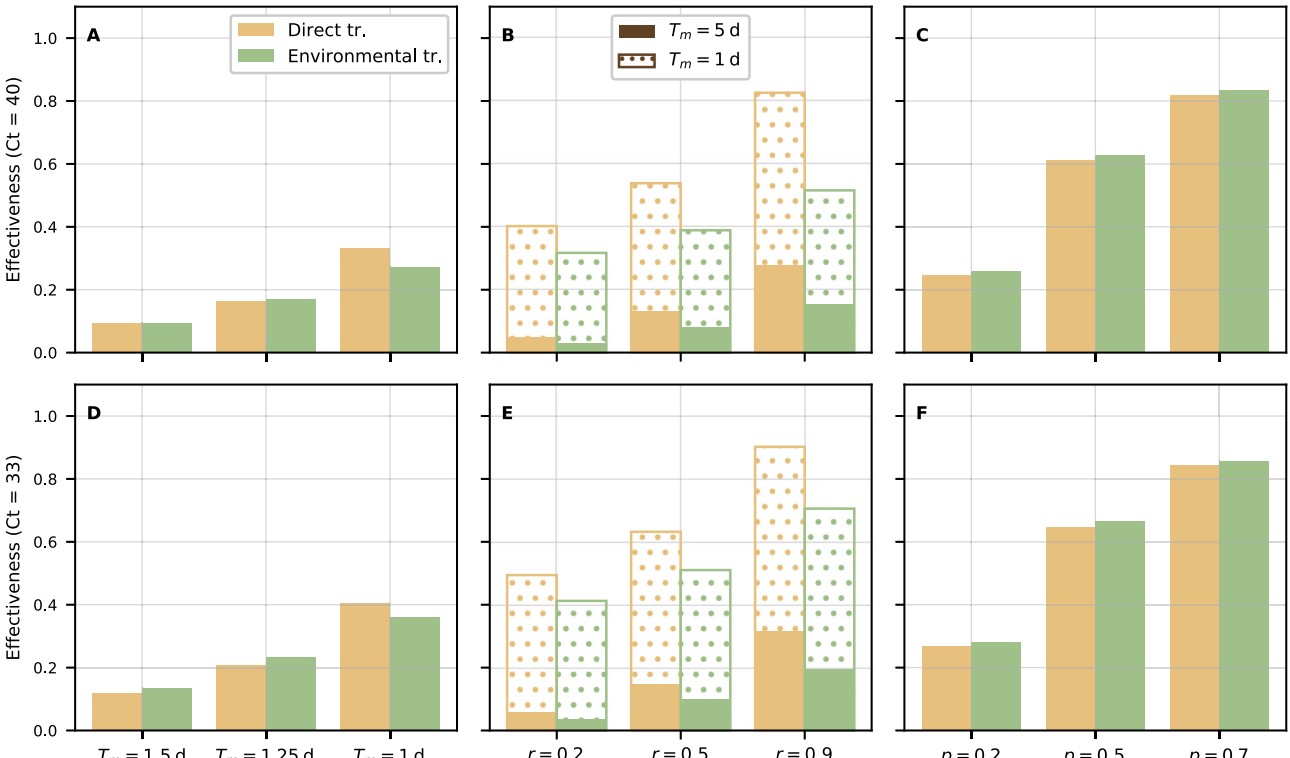

**Fig. 2 | Effectiveness of intervention measures.** Results for early removal/culling of unsold chickens (**A**, **D**), control of chickens entering the market (**B**, **E**) and preemptive immunisation through vaccination (**C**, **F**). Bars represent a mean reduction in average, cumulative daily prevalence with respect to a baseline scenario with no intervention, based on 5000 simulations from 500 posterior samples. The yellow and green bars correspond to direct and environmental transmission, respectively. In the latter case, we set $\Theta^{-1} = 3$ days for the sake of visualisation. In (**B**, **E**), solid and hatched bars correspond to a maximum length of stay of 5 (baseline) and 1 days, respectively. The first and second rows are based on posterior distributions obtained from fits to $Ct = 40$ and $Ct = 33$ data, respectively.

conditions (see Supplementary Methods). Within LBMs, the infection can be transmitted through poultry drinking water, as well as cages and floors that are contaminated by faecal material and during slaughtering. The lack of disinfection and the constant moving of cages and birds further promote poultry exposure to environmental contamination. Supplementary Fig. 10 shows a numerical validation of the mapping expressed in Eq. (1).

To start with, we implemented three measures based on either (i) early removal/culling of unsold chickens, (ii) control of chickens entering the market or (iii) preemptive immunisation through vaccination. Figure 2 displays the effectiveness of various interventions, computed as the reduction in cumulative daily prevalence relative to a baseline scenario with no intervention (See Supplementary Figs. 11 and 12 for prevalence dynamics over a single day). The green and yellow bars correspond to direct and environmental transmission, respectively. In the latter case, we present a single value of $\Theta$, but our results are independent of this choice. Estimates of intervention effectiveness presented in Fig. 2 are robust to mismatches between inferred and original parameters in simulated scenarios (Supplementary Fig. 13).

In (i), unsold chickens are automatically removed from the market if still unsold after a time $T_m$. Figure 2 shows that (i) is not effective at reducing prevalence (A, D), unless chickens are removed after 1 day or less. Indeed, high levels of transmission, combined with a short latent period in broilers (Fig. 1B), lead to a rapid build-up of infectious chickens well before $T_m$. This result holds, both qualitatively and quantitatively, regardless of whether we consider direct (green) or environmental (yellow) transmission.

Intervention (ii) aims at reducing the proportion of exposed chickens entering the market, either as the result of control measures

acting upstream, e.g. by enhancing farmers' and traders' compliance with biosecurity practices. In practice, we implement (ii) by reducing the proportion of previously exposed chickens from $\rho_{c,b}$ to $(1-r)\rho_{c,b}$, where $r$ represents the intervention's strength. Figure 2B, E reveal that a reduction in $\rho_{c,b}$ by a factor $r = 0.9$ alone (filled bars) is not sufficient to lower transmission significantly. Indeed, latent & infectious chickens arriving at the LBM, albeit fewer compared to baseline, are still able to sustain high levels of transmission. The effectiveness of (ii) is even smaller in the presence of environmental transmission due to AIV persistence in the environment, which is not directly affected by the intervention. However, a combined control strategy involving both (i) and (ii) proves superior to individual measures (hatched bars). Notably, the benefits of combining (i) and (ii) exceed expectations under the assumption that their effects were additive or multiplicative, hence suggesting a synergistic effect of multiple interventions (Supplementary Fig. 14).

With intervention (iii) a proportion $p$ of chickens are immunised through vaccination, and are assumed to be completely protected from AIV infection. This measure not only reduces the number of chickens entering the market while infectious or latent, but also reduces overall susceptibility to AIV in the flock. Figure 2C, F show that preemptive vaccination is particularly effective at reducing transmission; in particular, the reduction arising from vaccinating just 20% of all chickens is comparable to that of the most stringent implementations of interventions (i) or (ii).

The inclusion of environmental transmission in our model allowed us to explore the impact of sanitation, which is often adopted in the context of LBMs. Daily, weekly, or even monthly cleaning of poultry stalls, with or without weekly disinfection, was found to be associated with lower detection of AIV environmental

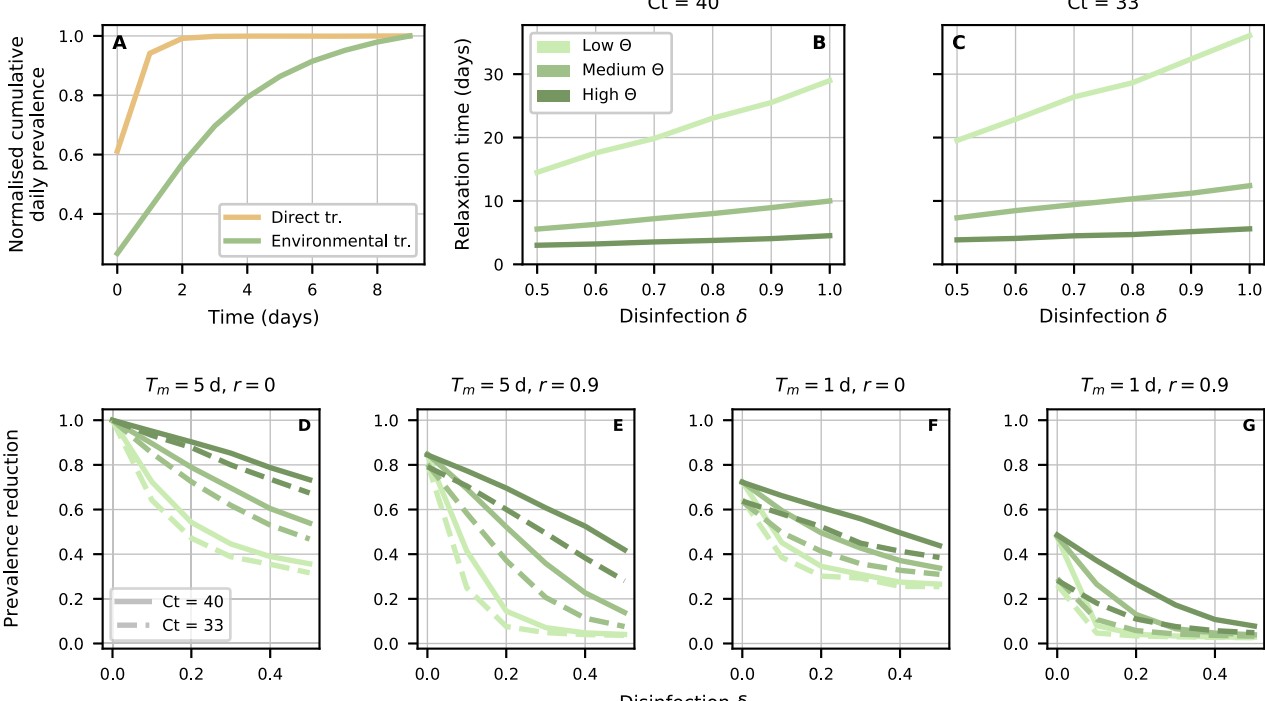

**Fig. 3 | Effectiveness of market depopulation and disinfection under direct vs environmental transmission. A** Cumulative daily prevalence, expressed as a fraction of its stationary value, after depopulating and fully disinfecting ($\delta = 1$) the LBM, under direct (yellow) and environmental (green) transmission. In the latter case, we set $\Theta^{-1} = 3$ days. **B, C** Average relaxation time as a function of disinfection $\delta$, based on $Ct = 40$ and $Ct = 33$ posterior distributions. Light to dark lines correspond to $\Theta^{-1} = 10, 3, 1$ days, respectively. Relaxation time is defined as the time at which cumulative daily prevalence crosses a given threshold value for the first time since LBM depopulation. Here, this threshold is set to a fraction (0.95) of the expected cumulative daily prevalence in the pre-intervention period. We compute 500 relaxation times from as many posterior samples, using 10 independent simulations to estimate mean cumulative daily prevalence. **D–G** Cumulative daily prevalence under various combinations of reduced length of stay (from left to right), reduced probability of prior exposure (from left to right) and disinfection, on the x-axis, for varying rates of environmental decay. Prevalence is calculated relative to a scenario with no interventions and the same $\Theta$. Results corresponding to solid and dashed lines are based on samples from $Ct = 40$ and $Ct = 33$ posterior distributions, respectively.

contamination[30]. However, sanitation is not straightforward to implement in practice[34,35]. Here, sanitation is assumed to reduce environmental contamination by a factor $\delta$. First, we note that while direct and environmental transmission were shown to yield similar stationary dynamics (Supplementary Fig. 10) and sensitivity to interventions (i) to (iii) (Fig. 2), significant dynamical differences arose in the presence of sanitation. Specifically, Fig. 3A shows that after depopulating and disinfecting the LBM, baseline prevalence levels were recovered rapidly under direct transmission, but not under environmental transmission. The mechanistic reason lies in the "inertia" inherent to the environmental reservoir, relative to an equivalent model with direct transmission. This inertia is expressed by the apparent trade-off between environmental transmissibility $\beta_{env}$ and persistence in the environment, as quantified by $\Theta$. We stress that while this effect follows from Eq. (1), it is not an artefact: $\beta_{env}$ and $\Theta$ should be expected to behave in this way, with, e.g., longer persistence in the environment (smaller $\Theta$) corresponding to slower relaxation. This is indeed confirmed by Fig. 3B, C, where we compare three values of $\Theta$ and use $Ct = 40$ and $Ct = 33$ posterior samples, respectively. At low $\Theta$, the typical relaxation time is at least 15 days and increases rapidly with disinfection $\delta$. As $\Theta$ increases, the relaxation time becomes shorter and less dependent on the disinfection rate.

Consistently with Fig. 3A–C, we found increasing returns from routinely (daily) disinfecting the market when $\Theta$ is small, even if disinfection is not perfect (Fig. 3D–G). A multi-pronged approach featuring interventions (i) and (ii) and small levels of disinfection, say $\delta = 0.3$, is able to curb cumulative daily prevalence by more than 80% for any explored value of $\Theta$ and in both parameter configurations

(Fig. 3G). Preventing 90% of prior infections (Fig. 3E) proved more effective than just limiting maximum length of stay to 1 day (Fig. 3F) when coupled with routine disinfection, but not in the absence of it (i.e. $\delta = 0$).

## Discussion

In this work, we characterised H9N2 transmission patterns in a single LBM in Bangladesh by fitting a mechanistic transmission model to a longitudinal dataset collected in the context of a field experiment.

Our results confirm the important role of LBMs as hotspots of AIV transmission. We found a high prevalence of H9N2 AIV, in agreement with previous studies and LBM surveillance in Bangladesh[15,16]. Our simulations further suggest that H9N2 AIV prevalence varies considerably during a single day due to high transmission rates. Such an effect has been illustrated in previous modelling work[22], and should be accounted for by AIV surveillance initiatives and in the design of chicken sampling strategies in general. From a systemic perspective, the high persistence and prevalence of H9N2 AIV in LBMs are concerning for the whole poultry production and distribution infrastructure in which LBMs are embedded. Although our analysis is based on data collected from a single LBM, our results are relevant to LBMs with similar features. Indeed, vendors operating in the same types of markets and locations are expected to adopt similar practices[20,25] and source chickens from overlapping catchment areas[20].

The fast turnover of susceptible chickens in LBMs is concerning since it is likely to promote amplification of AIV subtypes with short latency other than H9N2, e.g. H5N1 AIV[32]. This virus is routinely

detected in Bangladeshi wholesale markets, albeit at a lower frequency compared to H9N2 AIV[14]. This likely reflects the lower abundance of traded backyard ducks, which act as the primary source of H5N1 infections in markets[15,36].

We estimated an average latent period of 5.3–6 h and 1–1.3 days, depending on the $Ct$ threshold, for exotic broiler and backyard chickens, respectively. Short latent times in exotic broiler chickens are compatible with a fast onset of viral shedding, already after one-day post-inoculation, as observed in laboratory experiments[32,37–43]. Moreover, we believe that our experimental design, which includes inter-sampling periods as short as 12 h, is more suitable to resolve short latent periods than many laboratory experiments, which typically collect the first samples post-inoculation only after 1 day. Our estimates were robust with respect to prior assumptions about the duration of shedding, as shown in sensitivity analyses. Unfortunately, we could not reliably estimate the infectious period since our data did not include enough information about viral clearance.

Inferred proportions of chickens that were recruited directly in farms (intervention group) and that had already been exposed to H9N2 AIV prior to $T_0$ revealed substantial differences between broiler and backyard chickens. Specifically, we found most exposed broilers to be actively infected at recruitment, with little evidence of accrued immunity. In contrast, the majority of backyard chickens were estimated to be already immune to H9N2 AIV at recruitment. A recent study found 1% and 15.7% H9N2 AIV antibody prevalence and low viral prevalence, 0.2% and 0.5%, in broiler and backyard farms around Chattogram, respectively[44]. These prevalence values are slightly lower than estimates reported from active surveillance, which found 2.2% and 9.6% of AIV RT-PCR positivity in backyards and farms, respectively, with around a fourth of positive samples attributable to H9N2 AIV[14]. At the flock level, H9N2 AIV prevalence around Chattogram has been estimated at around 0.7% and 1.9% for backyard and broiler chickens, respectively. Another cross-sectional study of household chickens performed in the same area found a household-level prevalence of H9N2 AIV of 3.2% [45].

In absolute terms, our estimates of H9N2 AIV circulation in broilers sampled at $T_0$ are larger than previous estimates of viral circulation in farms. In fact, crude numbers of broiler chickens recruited in farms that tested positive for H9N2 AIV at $T_0$ (5 out of 110), suggest higher viral prevalence than found by other cross-sectional studies. Analogously, we estimated a higher proportion of past infections in backyard chickens at $T_0$ than suggested by serological evidence. While the reasons for these discrepancies remain unknown, we note that chickens included in this study were collected towards the end of a production cycle, when they might be exposed to an increased risk of AIV infection. Nonetheless, our results remain in broad qualitative agreement with available evidence as both suggest a higher prevalence of antibodies against H9N2 AIV in backyards compared to broiler farms, in the face of larger viral circulation in broilers.

Exotic broilers recruited at farm gates were found to be less likely to be already exposed to H9N2 AIV compared to chickens recruited at LBM gates (control group), suggesting some degree of viral amplification happening along channels connecting farms to markets[15,20]. However, we found the opposite relation in the case of backyard chickens. One possible explanation is that backyard farmers included in this study saw an opportunity to sell chickens that were already sick, potentially due to AIV infection. Selling sick birds is not an uncommon practice among backyard farmers near Chattogram, who often operate in a world of compromises[46].

High levels of H9N2 AIV circulation in LBMs are concerning from a veterinary public health standpoint, and may require considerable efforts and resources to be controlled effectively. Indeed, some of our simulated interventions, like reduced length of stay and reduced probability of prior exposure, proved to be only modestly effective.

Combining both interventions proved considerably more effective at reducing transmission compared to individual measures. Bans on overnight stays in Hong Kong were estimated to reduce H9N2 AIV isolation rates by more than 80%[23]. It is possible that the combination of high introduction levels and baseline within-market transmission is larger in our study, thus requiring increased efforts to reduce transmission by an amount similar to what had been observed in Hong Kong.

Preemptive vaccination alone proved to be particularly effective in simulations, under the assumption of complete sterilising immunity. A vaccine against H9N2 AIV is already available in Bangladesh, but its use has been limited to breeders and layers[47]. Widespread H9N2 AIV vaccination has been implemented in China and Korea. In Korea, the genetic diversity of H9N2 AIV decreased suddenly after implementing vaccination in 2007[48]. Large-scale AIV vaccination stamped out H7N9 in Chinese LBMs[49] but not H9N2, likely due to vaccine failure[50]. Indeed, continued AIV evolution can jeopardise vaccination efforts, requiring effective viral surveillance to inform vaccine composition and timely roll-out of updated vaccines.

We considered two alternative modes of transmission, direct and mediated by the environment. Both scenarios were able to explain observed dynamic patterns and yielded similar results in the context of interventions targeting chickens only. Previous theoretical work indeed demonstrated that both modes of transmission lead to similar dynamical outcomes, especially when environmental contamination unfolds on a fast time scale, and that it may be difficult to prefer one or another based solely on prevalence or incidence data[51,52]. This is a reassuring finding as it suggests that some epidemiological conclusions are not affected by precise modelling assumptions. However, further work is needed to identify dynamical signatures of direct and environmental transmission. Nonetheless, incorporating environmental transmission is necessary if the objective is to assess the impact of LBM depopulation and routine cleaning/disinfection, as done in this work. In this case, moderate levels of cleaning were able to curb transmission significantly in simulations, especially with small decay rates, as that corresponds to a slower accumulation of contaminated material. Periodic cleaning/disinfection, usually carried out during rest days, has been shown to reduce the AIV burden in Chinese, Hong Kong, and Bangladeshi LBMs[22,30,53]. Including environmental transmission may also be more appropriate to capture differences in the prevalence of contamination across LBM sections (e.g. stalls and slaughter areas)[15], and assess how the distance from slaughter areas affects the risk of contamination. Practical difficulties in successfully implementing sanitation in LBMs[35] further underscore the importance of adopting a multi-pronged approach to reduce the burden of H9N2 AIV in LBMs. Our study also makes the case for the vaccination of poultry intended to be sold in LBMs in Bangladesh.

Our study has several limitations. It focused on exotic broiler and backyard chickens, i.e. the same chicken types sampled in the field experiment. We did not include other chicken types, quails or ducks that are traded at the same market, as it would have been difficult to estimate additional parameters in the absence of appropriate data. While this could potentially bias our estimate of AIV transmissibility, which appears to be sensitive to other prior assumptions as well, we believe that our main results, e.g. estimated prevalence, are not affected by these simplifying study conditions. We did not consider seasonal variation in AIV transmission over the study period[54]. Nonetheless, explored contamination decay rate values can be sensibly mapped to environmental conditions at different times of the year.

We assumed that PCR tests could not detect infections during the latent phase, i.e. in absence of viral shedding, but were otherwise perfectly sensitive in the case of infectious and recently recovered chickens. High rates of positivity to H9N2 AIV suggest however that test sensitivity should not be a problem in our analysis. We also believe

that positive outcomes were unlikely to arise from cross-reactivity induced by other AIVs, but we can not exclude cross-contamination of some samples in the laboratory. We note that immune cross-reactions between distinct AIVs may still affect susceptibility to H9N2 AIV. In addition, it has been proposed that backyard chickens are intrinsically more resistant to AIV infection compared to exotic broilers[55–58], which could partially explain differences in attack rates between them. Our results indicate, however, that differences in earlier exposure to H9N2 AIV are sufficient to explain observed incidence patterns, and are consistent with known ages at sale and levels of H9 seropositivity in broilers and backyard chickens[44]. A better understanding of the effectiveness of prior immunity, which we have assumed to be sterilising, will help validating or confuting this interpretation.

In conclusion, we found that H9N2 AIV is transmitted rapidly among chickens in an LBM in Chattogram, Bangladesh. A short latent period, especially in broilers, high transmission rates and a continuous daily supply of susceptible chickens provide fertile grounds for H9N2 AIV amplification despite the short length of stay. Virus persistence in LBMs is further promoted by poor cleaning, which enables viral accumulation in the environment, and frequent introductions of infectious chickens from trade. Consequently, sustained efforts involving a diverse range of veterinary public health interventions will be required to curb the circulation of this virus. The ubiquity of similar poultry handling and trading practices suggests that our findings apply to other Bangladeshi LBMs as well. Applications of the model to other LBMs may require calibrating specific LBM features such as the number of chickens being traded and their length of stay. Applications to other AIV strains, e.g. H5N1, will also require accounting for the presence in the LBM of relevant hosts species (e.g. waterfowl in the case of H5N1 AIV), their specific infection parameters and differential ability to survive in the environment[59]. Finally, we note that our modelling framework could be applied to disentangle the contributions of external introductions and local transmission in other types of live animal markets and host-pathogen systems.

## Methods
### Model description
We use a SEEIRR model to simulate disease dynamics. This model is a variant of the more common SEIR model, which is typically used to investigate AIV dynamics[60]. Under the assumptions of density-dependent transmission and homogeneous mixing, susceptible ($S$) chickens become infected at rate $\Lambda(t) = \beta I(t)/N$, where $\beta$ is the transmission rate, $I(t)$ counts the number of infectious ($I$) chickens at time $t$ and $N$ is the number of new chickens entering the market daily. Exposed ($E$) chickens turn infectious after an average latent period $T_E = \sigma^{-1}$ and recover after an average infectious period $T_I = \mu^{-1}$. The exposed state consists of two consecutive stages ($E_{1,2}$) with the same exit rate $2\sigma$, yielding a gamma-distributed latent period. This is often regarded as a more realistic assumption than an exponential distribution of durations, which is implicit in models with single-staged compartments[32]. On the other hand, including a single infectious compartment should not affect our results significantly since the main limitation to transmission within LBMs comes from the short length of stay. Recovered chickens initially enter the $R_+$ state and then advance to $R_-$ at rate $\eta$. In this work, we assume that only biological samples retrieved from $I$ or $R_+$ chickens can yield a positive PCR test result. The distinction between $R^+$ and $R^-$ compartments allows us to capture the persistence of viral RNA in infected chickens that recently stopped shedding[61]. We assume that the two chicken types considered here, exotic broiler and backyard chickens, share the same biological parameters, except the latent period.

We model an open population of chickens that mimics the activity of an LBM: more in detail, we assume that $N_b$ new chickens of type $b = BR, BY$ reach the market in bulk every day, always at the same time (note that $N = \sum_b N_b$). Of these, a proportion $\rho_b$ has already been exposed to influenza prior to entering the market. Chickens are then sold progressively over time, their length of stay being distributed as in Supplementary Fig. 1A. We assume for simplicity that the distribution of length of stay of backyard chickens is the same as that of broilers. Supplementary Fig. 15 shows that this assumption does not affect epidemic dynamics significantly.

### Equivalence between direct and environmental transmission
Under environmental transmission, the expression for the force of infection becomes $\Lambda_{env}(t) = \beta_{env} I_{env}(t)/N$, where $I_{env}(t)$ represents viral load in the environment at time $t$; its physical units are arbitrary, but chosen in a way that $I_{env}$ increases by an amount $I(t)$ (i.e. the prevalence of infectious chickens) between $t$ and $t+1$.

A mapping between $\beta$ and $\beta_{env}$ that (approximately) preserves stationary viral dynamics can be obtained as follows: let $\tilde{T}$ denote the average time a single chicken spends at the market while infectious. Under direct transmission, its spreading potential is given by $\beta\tilde{T}$; under environmental transmission, the same quantity is evaluated as:

$$\beta_{env}\tilde{T}\sum_{t=0}^{\infty}e^{-\Theta t}, \tag{2}$$

where the last sum accounts for the persistence and progressive decay of infectious faeces in the environment. Equating the two expressions yields the relation $\beta_{env} = \beta \cdot (1 - e^{-\Theta})$.

### Field data collection
The field experiment consisted in caging 10 chickens together at a market stall for 84 h, and sampling them for positivity to AIV at four time points, $T_1 = 0$, $T_2 = 12$, $T_3 = 36$ and $T_4 = 84$ h during the duration of the experiment. Of these 10 chickens, a group of 5 were recruited directly at the market right before $T_1$ (control group), while the remaining 5 birds had been recruited 2.5 days in advance ($T_0$) from farms (intervention group) and stored in a biosecure environment before being introduced to the LBM at $T_1$. The experiment was repeated 30 times with exotic broilers and 34 with backyard chickens for a total of 300 and 340 chickens, respectively. In this work, we removed 80 broiler chickens corresponding to 8 experimental replicates where there was a suspect of cross-contamination of samples. More details about the experimental design can be found in ref. 33. Ethical approval for the initial study was obtained from both City University of Hong Kong and Chattogram Veterinary and Animal Sciences University.

### Fitting the model to field data
In the context of experimental data, we further distinguish between intervention ($i$) and control ($c$) chickens. This translates into four introduction parameters $\rho_{g,b}$, according to each combination of group $g \in \{c, i\}$ and type $b$. We assume that control and bulk (i.e. marketed chickens that were not part of the experiment) chickens are equivalent in all aspects, meaning that $\rho_b = \rho_{c,b}$. Finally, compartment-specific introduction probabilities are fully determined by specifying three hyper-parameters $\lambda_{BR}$, $\lambda_{BY}$ and $\kappa$. Briefly, these set the timing of prior exposure, under the assumption that the latter is gamma-distributed with type-specific rate $\lambda$ and shared shape parameter $\kappa$. Further mathematical details can be found in Supplementary Methods.

We used a Bayesian MCMC approach to infer parameters $\theta$ listed in Table 1. We chose priors that penalise large values of $\beta$ and set a narrow range for $T_{EI} = (\sigma_{BR}^{-1} + \sigma_{BY}^{-1})/2 + \mu^{-1}$, i.e. the average time from exposure to viral clearance; for a full account of fitted parameters' priors see Supplementary Table 1. The likelihood function is multinomial (see Supplementary Methods), and depends on the probability of a chicken testing positive for the first time at market entrance, i.e. $T_0$ or $T_1$, or during any other time segment $[T_j, T_{j+1}]$; in addition, we also account for chickens that remain susceptible throughout the experiment or until early removal. We resort to numerical simulations to

evaluate the likelihood, since an explicit representation of individual probabilities in terms of model parameters is not available. Simulations feature both bulk and recruited chickens from intervention and control groups. Importantly, we assume that recruited chickens do not contribute to transmission, but they can still be affected by exposure to infectious bulk chickens, which are way more abundant than the former. Intervention and control animals are recruited at times $T_0$ and $T_1$, respectively, and can not leave the market. From $T_0$ to $T_1$, intervention chickens are completely isolated from any source of infection, consistently with experimental conditions.

The inference routine is based on an ensemble sampler from the Python module *emcee*, version 3.1.1[62]. Briefly, this sampler runs $l$ chains in parallel, and makes proposals based on the collective state of all chains. We checked MCMC convergence by visual inspection, e.g. by looking at trace plots (Supplementary Fig. 16), and by looking at MCMC acceptance rates.

### Reporting summary
Further information on research design is available in the Nature Portfolio Reporting Summary linked to this article.

## Data availability
Field experiment data was collected in a previous study[33]. The raw data can be accessed on Zenodo[63].

## Code availability
Source code necessary to reproduce the analysis can be accessed on Zenodo[63].

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

## Acknowledgements

F.P., S.G., M.A.H., R.M., P.W., D.P. and G.F. are supported by the UKRI GCRF One Health Poultry Hub (Grant No. BB/S011269/1), one of twelve interdisciplinary research hubs funded under the UK government's Grand Challenge Research Fund Interdisciplinary Research Hub initiative. G.F. is supported by the French National Research Agency and the French Ministry of Higher Education and Research. The authors would like to acknowledge the use of the University of Oxford Advanced Research Computing (ARC) facility in carrying out this work.

## Author contributions

Conception and design: F.P., G.F.; Data Collection: L.K., D.P., G.F.; Data Analysis: F.P.; Methodology: F.P.; Investigations: F.P., L.K., J.L., S.G., Md.A.H., R.M., P.B., D.P., G.F.; Visualisation: F.P.; Writing–original draft preparation: F.P., G.F.

## Competing interests

The authors declare no competing interests.
