## [Peer Review File · Nature Communications]

Modelling the transmission dynamics of H9N2 avian influenza viruses in a live bird marketREVIEWER COMMENTS

Reviewer #1 (Remarks to the Author):

This paper analyses live bird market data on the transmission of H9N2 (low pathogenicity) avian influenza, fitting a 13 parameter model to field data from an LBM in Cambodia.

The paper is broadly speaking clearly written for a modelling heavy paper. It clearly goes through model parameters and discusses in detail the results of interventions from the fitted model. The supplementary information contains considerable detail on the model fit itself and justification for priors. The illustration of fit to simulated data is welcome. On a related point, while the fit to simulated data is helpful, it does show that some parameters are poorly recovered in the posterior. This may not be important in terms of the overall paper conclusions but the authors need to show this. One thing that might help, is to compare the results of interventions for the original simulated parameters, and the results using the recovered posteriors. If key outcomes are insensitive to this difference, it provides some confidence that those differences posteriors are not important. Unfortunately it is at best partial confidence of course (not accounting for differences between the model and the real world processes behind the data of course) but would be an important step.

Another thing makes the paper difficult to evaluate - nowhere did I see the actual fit of the model to the data; we have some evidence that we have the best fitted model, but we don't have much evidence on how good the best fit is.

In addition, a discussion of the fundamental drivers of the transmission dynamics (what is R_0 ? $R(t)$? the generation time of transmission? What proportion of susceptibles are infected in the first, second or third generation)? What are the initial conditions and how much of infection is due to introduction rather than within market transmission. These are important because they help us to understand the "why's" of the interventions.

A broader point is that the motivation behind the analysis seems to sit a bit between two stools. High prevalence of H9N2 in markets and the issue of control may be important for one of two reasons - first, if it results in substantial zoonotic infection, or if it is important for maintaining the circulation of the virus. However, H9N2, even though it may contribute to the generation of zoonotic forms of flu, is itself not a zoonosis. In this case, the prevalence in the market is less important than the role the market has in the circulation of H9N2 - and this can only be evaluated by considering onward transmission from the market. The authors need to make a better case for why their analysis helps us to understand or control that circulation of virus. This is discussed in the introduction (Paragraph starting line 49) but this at best touches upon the circulation question. There is more detail about trading networks in their other papers (e.g. Moya et al 2021) that they reference, so it would be useful to contextualise the analysis of this LBM in that study (e.g. is the LBM an important source of birds for mobile traders?).

A few minor points follow:

line 41: "especially"

line 81: this reference is a preprint which is useful to have but it would be better if, the interim has it been submitted for publication and/or accepted.

line 97: I struggled a bit with the definition of control and intervention chickens. It's clearer when you look at the companion preprint (Kohnle et al) but a bit more here would be useful.

line 123: this is a striking result - are these all infected in one generation due to a high prevalence of infection of birds entering the market? Or is there another reason?

line 139: why "tentative"

line 170 onwards. As discussed, it seems that in the model, environmental contamination only differs from direct transmission by allowing for a longer term persistence once infected birds are

gone. Otherwise the mechanism of transmission (and impact on transmission) is the same. Kim et al 2018 which the authors cite suggest that there is a strong variability in prevalence across stalls and slaughter areas of the market - is this likely to make a difference here in terms of the way contamination vs direct transmission would work?

line 220 onwards - are there any references that would help us to understand how difficult it would be to sanitise market? A brief description of what this would entail would be helpful.

line 227: there is a useful point here in terms of the recovery of infection - and it would therefore be important to gain a better understanding of why - referring to my earlier general point, is it because, for example the R value is very high? Or is it because the incoming infection prevalence is high.

line 342: That the two scenarios of transmission yield similar result is reassuring but as noted earlier, I suspect this may at least partially be because the dynamics in the model really aren't all that different. While the authors cannot be expected to do everything in a single paper, I think some discussion of the implications of model similarity is worthwhile.

line 361: "we believe that our main results ... are not affected ..." - why?

line 381: "with similar conditions" - which are the key conditions that are important?

S.92 onwards. after all long discussion of the effects of environmental conditions on environmental survival, on line S.103 the authors then reduce the difference to one of temperature only.

supp fig 7 - as discussed at the beginning of this review, really useful to have this but more details on the parameter sets - were they chosen to be relevant to the posteriors for the real data (e.g. drawn from the posterior distributions)? Also, the mismatch in scenario 5 is indeed striking and it would be good to have confidence it isn't important.

Reviewer #2 (Remarks to the Author):

The authors present an interesting study modelling the transmission dynamics of avian influenza with parameters fit to empirical data from a live bird market. I find the model proposed to be reasonable; the statistical analysis appropriate; and the limitations of the approach to be unambiguously communicated. Generally, I find the manuscript to be uncommonly clearly written.

Regarding the presentation of the model, I would encourage the authors to briefly state (perhaps any additional analysis is beyond the scope of this manuscript) how they would expect compartment topology to modify the parameter estimates or otherwise invest more time in the model description to motivate why SEIIR is clearly the most appropriate framework.

The authors describe several predicted epidemiological differences between exotic broilers and backyard chickens including:

"From our model's output, we found a shorter latent period in exotic broiler compared to backyard chickens (Fig. 1B,C), lasting an average of 5.3 hours for exotic broiler, and 1 days for backyard chickens."

and

"in the case of exotic broilers, most chickens with prior exposure to H9N2 were either infectious or latent, with only a minor proportion of them being immune (Fig. 1E). In contrast, most previously-exposed backyard chickens were immune to H9N2 (Fig. 1F). Our results thus tentatively suggest that prior infection occurs close to marketing age for broilers, whereas in backyard chickens it may occur further in the past, which is consistent with the latter being raised for a longer time compared to broilers."

I encourage the authors to comment on whether they believe these findings are entirely consistent with variable patterns of prior exposure for these two populations or if the data presented suggest that there may be some unrecognized genetic determinants of susceptibility. Additionally I understand that to support the second passage quoted above, there has been an underlying assumption made that prior infection confers long lasting immunity (relative to the life cycle of a market chicken). If this is correct, I encourage the authors to explicitly state this assumption and perhaps introduce some additional background information regarding the market chicken life cycle so that the unfamiliar reader, like myself, may have some expectation of the maximum number of infections for one individual.

The authors consider both direct and environmental transmission, writing, "Ienv(t) accumulates due to shedding from infectious chickens and decays progressively at rate Θ . Here we consider three values of Θ , namely $\Theta-1 = 10, 3, 1$ days, corresponding to slow, intermediate and fast decay, respectively. These values are based on actual estimates from the scientific literature and capture a broad range of environmental conditions." I encourage the authors to briefly describe the physical nature of environmental transmission in a live bird market for readers, like myself, who are more familiar with influenza transmission dynamics in the human population which I expect substantially differ.

I find one of the more striking findings reported in the manuscript to be that while, "reduced length of stay and reduced probability of prior exposure, proved to be only modestly effective. Combining both interventions proved considerably more effective at reducing transmission compared to individual measures" I encourage the authors to clarify in the text if this effect is in fact nonadditive and if so to include some description of the expected mechanism behind the non-additivity.

In conclusion, the authors present an interesting and well written study which I believe will be of broad interest.

Sincerely,
Nash Rochman (invited 12/5/23, returned 12/19/23)

Reviewer #3 (Remarks to the Author):

The manuscript titled "Modelling the transmission dynamics of H9N2 avian influenza viruses in a live bird market" aims to understand the transmission dynamics of H9N2 avian influenza viruses in live bird markets (LBMs) in Bangladesh. The study uses a mechanistic transmission model, fitted to field data collected in a specific LBM, to simulate the spread of the virus among poultry.

The study presents significant findings on H9N2 transmission in LBMs, highlighting a rapid infection rate among susceptible chickens, often within a day. The virus's latent period is notably short, particularly in broiler chickens, sometimes as brief as 5.3 hours, facilitating its swift spread. It also uncovers considerable differences in exposure and immunity between exotic broiler and backyard chickens. Its unique approach, combining field data with mathematical modelling, and the robustness of its findings, underscored by sensitivity analyses and Bayesian methods, mark it as particularly relevant for areas where H9N2 is prevalent, guiding interventions to control its spread. Thus, This research offers crucial insights into the epidemiology of H9N2 in LBMs, vital for public health and the poultry industry.

I thoroughly enjoyed reading this manuscript and believe it is already of high quality. However, I suggest incorporating discussions on several interesting points. Firstly, the study's results are drawn from a single LBM in Bangladesh. It would be beneficial to explore how these findings might apply to other LBMs with varying dynamics, bird populations, and environments. Additionally, a more in-depth examination of environmental factors, like temperature, humidity, and market hygiene, is crucial as they significantly influence H9N2's survival and spread. A comparison with other avian influenza strains could also offer a broader perspective, highlighting differences and

similarities in transmission dynamics. Finally, it would be interesting to consider whether these findings could be applicable to livestock markets other than those for birds and discuss the generalizability of such high infection rates in group settings, dependent on the virus.

General information

First of all, we thank the three anonymous reviewers for their assessments and helpful comments. Please find our detailed replies to these comments below. We also provided a copy of the revised manuscript and supplementary material where all the main changes have been highlighted in red/blue.

Reviewer 1

This paper analyses live bird market data on the transmission of H9N2 (low pathogenicity) avian influenza, fitting a 13 parameter model to field data from an LBM in Cambodia.

The paper is broadly speaking clearly written for a modelling heavy paper. It clearly goes through model parameters and discusses in detail the results of interventions from the fitted model. The supplementary information contains considerable detail on the model fit itself and justification for priors. The illustration of fit to simulated data is welcome. On a related point, while the fit to simulated data is helpful, it does show that some parameters are poorly recovered in the posterior. This may not be important in terms of the overall paper conclusions but the authors need to show this. One thing that might help, is to compare the results of interventions for the original simulated parameters, and the results using the recovered posteriors. If key outcomes are insensitive to this difference, it provides some confidence that those differences posteriors are not important. Unfortunately it is at best partial confidence of course (not accounting for differences between the model and the real world processes behind the data of course) but would be an important step.

We are pleased that the reviewer appreciated our work and we thank them for their comments.

The reviewer raises an important point, and we agree that further work is required to assess to what extent the discrepancies between "true" and recovered parameters affect other key findings. Following the advice from the reviewer, we assessed the effectiveness of individual interventions displayed in Fig. 2 using original parameters and recovered posteriors from the scenarios reported in Supplementary Fig. 7 (now 9). These results are displayed in the new Supplementary Fig. 13 (attached below). Each column in the latter figure represents a different scenario (i.e. simulated dataset), while rows correspond to distinct interventions. We find an excellent agreement between simulations using original (dots) and recovered (bars) parameters in terms of intervention effectiveness in the case of:

- Shorter length of stay (first row)
- Shorter length of stay + Control of chickens entering the market (third row)
- Vaccination (last row)

However, we also underestimate the effectiveness of reducing past exposure in chickens entering the market in scenarios 2 and 3 (second row, second and third columns). As a

reminder, the latter intervention reduces the probability that an incoming chicken is not susceptible by a factor $1 - r$. Upon further investigation, we discover that such discrepancy disappears completely in scenario 2 and is considerably reduced in scenario 3 after setting the probability $P_{c,b,X}^{(0)}$ that an incoming bulk (i.e. control group) chicken of type b is in compartment X to the corresponding true value. These results are shown in Fig. A below alongside this reply for completeness.

Overall, these findings suggest that our main results about the effectiveness of modelled interventions are robust to modest discrepancies in posterior estimates. The new supplementary figure is referenced in the Results section:

"Estimates of intervention effectiveness presented in Fig. 2 are robust to mismatches between inferred and original parameters in simulated scenarios (Supplementary Fig. 13)."

Supplementary Figure 13: Estimating effectiveness of interventions in artificial scenarios. This figure compares the effectiveness of veterinary public health interventions calculated using recovered (bars) and original (dots) parameters in the 5 artificial scenarios considered in Supplementary Fig. 9 (from left to right). We consider the same interventions presented in Fig. 2 in the main manuscript: shorter length of stay (top row), control of infected chickens entering the LBM (second row) and vaccination (bottom row). The third row corresponds to the simultaneous implementation of the first two interventions. Moderate bias in posterior estimates of model parameters (Supplementary Fig. 9) does not alter our conclusions about the impact of interventions. Panels G,H indicate an underestimation of the impact of controlling incoming infected birds, which we ascribe to overestimating $p_{c,b}$ in artificial scenarios 2 and 3. Results are based on 1000 and 5000 simulations for original and recovered parameters, respectively. In the second case, simulations are evenly split across 500 independent samples from the recovered posterior distribution.

Fig. A: Re-analysis of panels G, H in Supplementary Fig. 13.

Another thing makes the paper difficult to evaluate - nowhere did I see the actual fit of the model to the data; we have some evidence that we have the best fitted model, but we don't have much evidence on how good the best fit is.

We agree with the reviewer about the importance of showcasing goodness of fit. In presenting our results, posterior predictive checks for the model fitted to Ct=40 data ended up in Supplementary Fig. 4 and 5 (now 7 and 8), while goodness of fit for the model fitted to Ct=33 data was not shown.

We now display posterior predictive checks for both fits to Ct=40 and Ct=33 data in Supplementary Fig. 3. The latter shows posterior predictions (alongside 95% C.I.) for the counts of chickens testing positive at different time points of the experiment (T0-T4, solid bars) or testing negative throughout the experiment (hatched bars). These predictions are further stratified by chicken type and recruitment group, and are overlaid to data (black dots).

Supplementary Fig. 3 suggests that the fit of the model to available data is adequate.

Supplementary Figure 3: Goodness of fit. Panels show posterior predictive checks for fits to Ct = 40 (A-D) and Ct = 33 (E-H) data, broken down by chicken type and recruitment group. Each panel shows posterior mean (bars) and 95% C.I. (grey error bars) for the numbers of chickens becoming positive (filled) at different stages of the experiment or remaining susceptible (hatched). Black dots and error bars denote data and 95% C.I. computed under

a binomial distribution assumption. We ran 20000 simulations with 5 control and 5 intervention chickens from 2000 independent posterior samples to estimate the probability of turning positive at different times. Expected counts were obtained by multiplying these probabilities by the number of recruited chickens in each category.

In addition, a discussion of the fundamental drivers of the transmission dynamics (what is R_0 ? $R(t)$? the generation time of transmission? What proportion of susceptibles are infected in the first, second or third generation)? What are the initial conditions and how much of infection is due to introduction rather than within market transmission. These are important because they help us to understand the "why's" of the interventions.

We thank the reviewer for this comment. In order to better disentangle the roles of external introductions and local transmission on H9N2 AIV spread, we measured additional epidemiological metrics in our simulations (using posterior samples to construct their posterior distribution).

First, we estimated the basic reproductive number R_0 for this model by considering an effective constant removal rate $\nu = T_{stay}^{-1}$, where T_{stay} is the average length of stay (obtained from the distribution of length of stay in Supplementary Fig. 1). With this simplifying assumption, R_0 can be calculated explicitly as:

$$R_0 = \sum_{b=BR,BY} \left(\frac{2\sigma_b}{2\sigma_b + \nu} \right)^2 \frac{\beta}{\mu + \nu} \frac{\bar{N}_b}{N} ,$$

where $N = \sum_{b=BR,BY} N_b$ is the total number of chickens introduced daily and \bar{N}_b is the average number of chickens of type $b = BR, BY$ present in the LBM. The expression above can be derived by means of the Next-Generation-Matrix for a multi-type $SEIR$ model (Diekman et Al., *Journal of the Royal Society Interface*, 2010; Champredon et Al., *SIAM Journal on Applied Mathematics*, 2018). The term \bar{N}_b reflects the assumption of density-dependent transmission and is calculated through simulations. This term also accounts for daily variations in population size due to trade (D'Onofrio, *Mathematical Biosciences*, 2002). Please note that the presence of N is due to our definition of the force of infection, where it appeared as a (constant) scaling factor.

The posterior distribution for R_0 is shown in Supplementary Fig. 6A. We find that this quantity is significantly larger than 1 (between 3.5 and 5 depending on Ct threshold), in agreement with our previous assessment of high spreading potential. It also provides a quantitative justification for the high probability of persistence (≈ 1) shown in the old Supplementary Fig. 6. The latter probability is now shown in Supplementary Fig. 6C against R_0 , which we vary through the transmission rate β . As expected, persistence is essentially 0 for $R_0 < 1$, and increases with R_0 as $R_0 > 1$. We also provide an estimate of

the basic reproductive number $R_0(\text{closed}) = \beta/\mu$ for a closed population ($v = 0$) with constant size N . As shown in Supplementary Fig. 6, $R_0(\text{closed})$ is extremely high, thus demonstrating the high infection pressure exerted on marketed chickens. Please note that we present $R_0(\text{closed})$ just for illustrative purposes, as the relevant transmission potential is quantified by R_0 . All new references have been added to the list of supplementary references.

Supplementary Fig. 6: Basic reproductive number and AIV persistence. (A) Posterior distribution for the basic reproductive number R_0 for fits to $Ct = 40$ (red) and $Ct = 33$ (teal). The latter is calculated as the basic reproductive number for a continuous time SEEIR model with a constant rate of chicken removal v , where v is set to the inverse mean length of stay. Combining previous theoretical results yields the expression

$$R_0 = \sum_{b=BR,BY} \left(\frac{2\sigma_b}{2\sigma_b + v} \right)^2 \frac{\beta}{\mu + v} \frac{\bar{N}_b}{N} \quad (\text{D'Onofrio, Mathematical Biosciences, 2002; Diekmann et Al., Journal of the Royal Society Interface, 2010; Champredon et Al., SIAM Journal on Applied Mathematics, 2018}),$$

where \bar{N}_b is the mean number of chickens of type b present in the LBM, and is estimated through simulations, and $N = \sum_b N_b$ is the total number of chickens introduced daily. (B) Posterior distribution for the ratio β/μ , which represents the basic reproductive number in a closed population ($v = 0$) of constant size N . (C) Posterior probability of AIV persistence as a function of R_0 . This is measured as the proportion of 2000 simulations where at least one latent or infectious chicken is observed at $t = 50$ days, assuming that all chickens entering the market after $t = 20$ days are susceptible. Results in A,B are based on 5000 posterior samples.

Next, we further characterised the relative contributions of introductions and secondary infections within LBMs, the effective reproductive number R_t and the mean generation time GT_t at different times of the day. These results can be found in Supplementary Fig. 5 and are summarised below:

- Panel A shows the proportions of chickens that enter the market as susceptible (S) and become infected (i.e. sold as E, I and R), or infectious (sold as I or R) before being sold. These proportions are further broken down by chicken type. We find that >70% of initially susceptible chickens are infected before being sold. More than 50% of broiler chickens are also able to become infectious before leaving the market due to their short latent period. In contrast, less than 25% of initially susceptible backyard chickens transition from E to I within the market.

- Panel B shows the proportion of infection events that were generated by chickens infected in the market rather than elsewhere prior to being offered for sale. While the posterior distribution for this quantity is rather wide, it shows that, on average, more than 50% of infections are caused by secondary, market-acquired cases.
- Panels C-F show R_t (C,E) and the mean generation time GT_t (D,F). To measure these quantities, we casted our model in an individual-based framework and tracked all transmission pairs; we then followed *Liu et Al., PNAS, 2018* and calculated R_t and GT_t as the mean number and mean generation time of infections generated by incident cases at time t (i.e. we used a "forward-looking" definition for both R_t and GT_t). Panels C,D are obtained under the same regime of external introductions considered in panels A,B and in the main analysis. Panels E,F, in contrast, correspond to the same scenario evaluated in Supplementary Fig. 6C, where all admitted chickens are susceptible, and all infections are locally acquired. In this case we find that viral circulation is sustained, i.e. the virus is able to circulate without becoming extinct. Panels C shows that the maxima of R_t correspond to the arrival of new chickens in the market (vertical dotted lines). R_t rises in the ~12 hours preceding a new shipment due to the sudden increased availability of susceptible hosts entailed by this event. Indeed, GT_t becomes shorter as a new shipment approaches. The spiky shape displayed by R_t within panel C follows from the observation that most infections occur when a new shipment arrives, i.e. when the size of the susceptible population is at its largest and readily infectious birds are admitted into the market.
- Removing introductions (panels E,F and Supplementary Fig. 6C) produces smoother R_t and GT_t curves that are not affected by the sudden introduction of infectious (I) birds. Panels E, F further reinforce our interpretation that $R_t > 1$ is driven by the cyclical enhanced availability of susceptible birds.

Overall, our additional analyses suggest that both external viral introductions and high within-market transmission are important drivers of H9N2 AIV epidemiology, as described in the main manuscript. These findings are mentioned in the Results section (below), and the figure is included in the Supplementary Material. These additional results are particularly relevant in the context of our main analysis, since they complement our previous findings and should hence be part of the supplementary material.

"The relative importance of external introductions of infected chickens and local transmission is assessed in Supplementary Fig. 5. We find comparable proportions of LBM-acquired infections caused by chickens infected before and after entering the LBM. Moreover, we estimate that within-LBM transmission is high enough ($R_0 = 3.7 - 4.9$ depending on Ct threshold) to ensure long-term persistence of H9N2 AIV within the LBM in the absence of further external introductions (Supplementary Fig. 6)."

Supplementary Figure 5: Relative importance of external introductions and local transmission. (A) Violin plots denote posterior distributions for the fraction of initially susceptible chickens that are infected locally and the fraction of those that become infectious before being sold. (B) Posterior distribution for the fraction of LBM-acquired infections that are caused by other chickens that became infected within the LBM, as opposed to externally-introduced infections. Results are based on 20000 simulations from 2000 independent samples from the posterior distribution. (C) Mean number of new cases (R_t) caused by chickens infected at time t . (D) Mean generation time versus the timing of primary infection t . The generation time is defined as the delay between the infection of a primary and a secondary case. Statistics in C,D were obtained by tracking transmission pairs in an individual-based version of our model. Panels E,F mirror C,D but assume that all chickens entering the market after 30 days are susceptible (i.e. in absence of external introductions). We consider all primary cases infected between $t = 45$ and $t = 48$ days. For externally-introduced infections, the infection time was set to the time of introduction. Daily shipments of chickens are denoted with vertical lines. Results in C-F based on 2000 simulations using the same number of independent samples from the posterior distribution.

A broader point is that the motivation behind the analysis seems to sit a bit between two stools. High prevalence of H9N2 in markets and the issue of control may be important for one of two reasons - first, if it results in substantial zoonotic infection, or if it is important for maintaining the circulation of the virus. However, H9N2, even though it

may contribute to the generation of zoonotic forms of flu, is itself not a zoonosis. In this case, the prevalence in the market is less important than the role the market has in the circulation of H9N2 - and this can only be evaluated by considering onward transmission from the market. The authors need to make a better case for why their analysis helps us to understand or control that circulation of virus. This is discussed in the introduction (Paragraph starting line 49) but this at best touches upon the circulation question. There is more detail about trading networks in their other papers (e.g. Moyen et al 2021) that they reference, so it would be useful to contextualise the analysis of this LBM in that study (e.g. is the LBM an important source of birds for mobile traders?).

We thank the reviewer for this comment. We strongly agree about the importance of embracing a systemic viewpoint in order to investigate pathogen spread across live poultry markets. In fact, we recently developed a computational model (<https://doi.org/10.1371/journal.pcbi.1011375>) to model AIV transmission within the entire poultry production and distribution network. While viral movements between LBMs represent a topic of crucial importance in general, *Moyen et Al.* found limited evidence of inter-LBM poultry movements in Chattogram. Nonetheless, onward spread from a LBM to others and farms could occur through the movements of traders' contaminated equipment and vehicles. We respectfully disagree with the reviewer about the nature of H9N2 AIV which, as stated in the Introduction, can be zoonotic (see also *Peacock et Al., Viruses, 2019* and *Peacock et Al., J. Virol., 2020*). It is then reasonable, in our opinion, to expect that larger prevalence increases the risk of AIV exposure for market workers and customers alike. Moreover, larger prevalence in poultry means more opportunities for viral reassortment and more exportations to other markets and farms.

A few minor points follow:

line 41: "especially"

This has been fixed.

line 81: this reference is a preprint which is useful to have but it would be better if, the interim has it been submitted for publication and/or accepted.

We confirm that the referenced preprint has been submitted for publication. However, we can not guarantee that this manuscript will be published/accepted in the short term as both manuscripts were submitted almost simultaneously. We commit to updating this reference as soon as we receive a notification of acceptance, conditional on the timeline of peer-review for the present work.

line 97: I struggled a bit with the definition of control and intervention chickens. Its clearer when you look at the companion preprint (Kohnle et al) but a bit more here would be useful.

We have clarified this distinction by adding further details. This sentence now reads:

"We further distinguished between chickens traded along conventional (control, c) and altered (intervention, i) marketing channels. The latter involved purchasing chickens from farms rather than from traders at the market, thus avoiding intermediate transport and storage steps."

line 123: this is a striking result - are these all infected in one generation due to a high prevalence of infection of birds entering the market? Or is there another reason?

This is one of our main results and follows from a combination of high transmission and high prevalence of infection in introduced chickens. The first aspect creates conditions to guarantee viral persistence in the market (see Supplementary Fig. 6), while the second aspect enhances the probability of infection due to the larger initial number of infections after each shipment. We believe that the new Supplementary Fig. 5 discussed above addresses the relative roles of transmission and introductions in an adequate manner. We emphasize that the survival curve shown in Fig. 1A is conditional on a chicken spending at least x hours in the market. In reality, only 11.5% of chickens remain in the market for more than 24 hours, the median length of stay being 16 hours (see Supplementary Fig. 1A).

line 139: why "tentative"

We removed this word.

line 170 onwards. As discussed, it seems that in the model, environmental contamination only differs from direct transmission by allowing for a longer term persistence once infected birds are gone. Otherwise the mechanism of transmission (and impact on transmission) is the same. Kim et al 2018 which the authors cite suggest that there is a strong variability in prevalence across stalls and slaughter areas of the market - is this likely to make a difference here in terms of the way contamination vs direct transmission would work?

The reviewer is right in pointing out that both modes of transmission lead to similar dynamical outcomes. In fact, previous theoretical work (*Cortez & Weitz, Am. Nat., 2013* and *Benson et Al., PLOS Comp. Biol., 2021*) demonstrates that direct transmission models can be seen as the limiting cases of indirect transmission models when the dynamics of environmental contamination is fast (as confirmed by our analysis as well). Some dynamical differences may arise in the case of slow environmental contamination, but *Cortez & Weitz* show that the specific dynamical signatures depend on the underlying model. On one hand, we believe that such similarity is reassuring, as it suggests that different modelling assumptions lead to similar conclusions about given epidemiological aspects. On the other hand, a realistic accounting of environmental contamination is necessary when addressing specific questions, such as assessing the impact of LBM depopulation and cleaning. Heterogeneities in the handling of meat and/or carcasses and cleaning practices across an LBM may lead to distinct outcomes if transmission is direct or indirect. Realistically, the latter mode should better capture differences in

prevalence of contamination across different market areas. For example, market stalls located near slaughter areas could be at an increased risk of contamination than more distant ones. Further discrepancies between models may arise in the case of frequent removal of diseased birds (as shown by *Benson et Al., PLOS Comp. Biol., 2021*), which we did not include in our model due to the low pathogenicity of H9N2 AIV.

We have now incorporated these points in the Discussion, and included the two references mentioned above. We also replaced one reference (*Martin et Al., Preventive Veterinary Medicine, 2011*) with a more comprehensive reference that reviews the impact of LBM disinfection and rest days (*Offeddu et Al., One Health, 2016*). The full paragraph discussing environmental contamination now reads:

"We considered two alternative modes of transmission, direct and mediated by the environment. Both scenarios were able to explain observed dynamic patterns and yielded similar results in the context of interventions targeting chickens only. Previous theoretical work indeed demonstrated that both modes of transmission lead to similar dynamical outcomes, especially when environmental contamination unfolds on a fast time scale, and that it may be difficult to prefer one or another based solely on prevalence or incidence data (Cortez & Weitz, Am. Nat., 2013; Benson et Al., PLOS Comp. Biol., 2021). This is a reassuring finding as it suggests that some epidemiological conclusions are not affected by precise modelling assumptions. However, further work is needed to identify dynamical signatures of direct and environmental transmission. Nonetheless, incorporating environmental transmission is necessary if the objective is to assess the impact of LBM depopulation and routine cleaning/disinfection, as done in this work.

In this case, moderate levels of cleaning were able to curb transmission significantly in simulations, especially with small decay rates, as that corresponds to a slower accumulation of contaminated material. Periodic cleaning/disinfection, usually carried out during rest days, has been shown to reduce AIV burden in Chinese, Hong Kong and Bangladeshi LBMs (Fournié et Al., Journal of the Royal Society Interface, 2011; Offeddu et Al., One Health, 2016; Chowdhury et Al., Emerging Infectious Diseases, 2020). Including environmental transmission may also be more appropriate to capture differences in prevalence of contamination across LBM sections (e.g. stalls and slaughter areas) (Kim et Al., Emerging Infectious Diseases, 2018), and assess how the distance from slaughter areas affects the risk of contamination. Practical difficulties in successfully implementing sanitation in LBMs (Barnett et Al., The Lancet Planetary Health, 2021) further underscore the importance of adopting a multi-pronged approach to reduce the burden of H9N2 AIV in LBMs. Our study also makes the case for the vaccination of poultry intended to be sold in LBMs in Bangladesh."

line 220 onwards - are there any references that would help us to understand how difficult it would be to sanitise market? A brief description of what this would entail would be helpful.

Chowdhury et Al., Emerg. Infect. Dis., 2020, which is already referenced in the manuscript, provides evidence of benefits of market cleaning even with monthly frequency in Bangladeshi LBMs. However, *Paritosh et Al. Transboundary and Emerging Diseases, 2017* found that a package of biosecurity measures revolving around cleaning did not yield differences in detection rates of H5N1 AIV. *Barnett et Al., The Lancet Planetary Health, 2021* suggest that the apparent limited effectiveness of LBM sanitation is due to practical implementation challenges, particularly the difficult social, economic and cultural context in which poultry workers operate. We tried to convey this nuance in the Results section when introducing disinfection (we have also included the new references in the main manuscript):

"Daily, weekly, or even monthly cleaning of poultry stalls, with or without weekly disinfection, was found to be associated with lower detection of AIV environmental contamination (Chowdhury et Al, 2019). However, sanitation is not straightforward to implement in practice (Paritosh et Al., Transboundary and Emerging Diseases, 2017; Barnett et Al., The Lancet Planetary Health, 2021)."

We believe that practical difficulties in achieving sanitation further underscore the importance of combining multiple interventions, which is one of our main conclusions. This point is now highlighted in the Discussion section as well:

"Practical difficulties in successfully implementing sanitation in LBMs (Barnett et Al., The Lancet Planetary Health, 2021) further underscore the importance of adopting a multi-pronged approach to reduce the burden of H9N2 AIV in LBMs."

line 227: there is a useful point here in terms of the recovery of infection - and it would therefore be important to gain a better understanding of why - referring to my earlier general point, is it because, for example the R value is very high? Or is it because the incoming infection prevalence is high.

We interpret the reviewer's comment as addressing the rapid recovery of infection following depopulation in the case of direct transmission rather than the difference between the latter and environmental transmission. As discussed above (see new Supplementary Fig. 5), fast transmission is due to the large transmission rate, while frequent introductions facilitate a quick return to endemic levels of prevalence. Indeed, if introductions were sporadic (e.g. not every day) it would then take longer to build up prevalence.

line 342: That the two scenarios of transmission yield similar result is reassuring but as noted earlier, I suspect this may at least partially be because the dynamics in the model really aren't all that different. While the authors cannot be expected to do everything in a single paper, I think some discussion of the implications of model similarity is worthwhile.

We already addressed this point above.

line 361: "we believe that our main results ... are not affected ..." - why?

We expect that including additional chicken types would have a small impact on our estimates since broiler and backyard chickens already represent a large proportion of poultry being traded in the LBM (Supplementary Fig. 1). Moreover, we know from additional field studies that prevalence in other chicken types is not higher than in broilers and backyard chickens (Kim et Al., *Emerging Infectious Diseases*, 2018). This suggests a smaller contribution of other chicken types to overall transmission compared to broilers.

line 381: "with similar conditions" - which are the key conditions that are important?

We slightly rewrote the final paragraph in the Discussion to expand on this point:

"The ubiquity of similar poultry handling and trading practices suggests that our findings apply to other Bangladeshi LBMs as well. Applications of the model to other LBMs may require calibrating specific LBM features such as the number of chickens being traded and their length of stay."

S.92 onwards. after all long discussion of the effects of environmental conditions on environmental survival, on line S.103 the authors then reduce the difference to one of temperature only.

The purpose of this section was to discuss potential factors affecting environmental survival while also presenting a broad range of values for the decay rate θ that were also realistic for this host-pathogen system. The sentence at line S103 just places the chosen values into the context of Handel et Al.'s findings (reference 2 in the supplementary), which include an explicit, AIV subtype-specific relationship between θ and temperature. We agree that this sentence may read confusing, and decided to remove it in the revised version of the Supplementary Material.

To further support the appropriateness of our choice of θ , we report various estimates of θ^{-1} from the literature as a function of water temperature (see Fig. B below). The relationship presented by Handel et Al. is shown in black, while individual red dots are distinct estimates retrieved from a systematic review. Importantly, these estimates were obtained under a variety of environmental conditions beyond temperature. Blue and orange lines correspond to values used in two distinct modelling studies investigating AIV transmission in poultry markets. We prefer to keep Fig. B in this letter as we believe that current references in the Supplementary Material already provide sufficient context.

Fig. B: Summary of estimates of θ^{-1} from the literature.

supp fig 7 - as discussed at the beginning of this review, really useful to have this but more details on the parameter sets - were they chosen to be relevant to the posteriors for the real data (e.g. drawn from the posterior distributions)? Also, the mismatch in scenario 5 is indeed striking and it would be good to have confidence it isn't important.

These parameter values were not selected from the posterior, although the scales of some of these were consistent with posterior estimates reported in the main manuscript. Our rationale was to verify not only whether we could recover initial parameters under optimal conditions (i.e. in absence of model misspecification), but also differences between chicken types and recruitment groups. This is now explained in the caption of Supplementary Fig. 7 (now 9)

The mismatch occurring in scenario 5 is evident mostly in the context of the duration of infection T_I . However, as we describe in the corresponding caption, this is partially intended since scenario 5 uses the same parameter set of scenario 4, but uses a misspecified prior on the total duration of shedding $T_E + T_I$. The observed mismatch arises because our data is mostly informative about T_E and not T_I , as recovery from infection is not tracked in our experimental setting. Please note that some degree of prior misspecification on $T_E + T_I$ is present also in scenarios 1-4, but is not as strong as in scenario 5. We have added a brief sentence in the caption of Supplementary Fig. 7 (now 9) to elucidate this point. We also note that while parameters describing the timing of prior infections (λ, κ) also show some mismatch, the mean and standard deviation of the corresponding distributions display a better agreement with the original values.

As discussed above, mismatches between original and recovered parameters did not significantly affect our assessment of effectiveness of interventions (see new Supplementary Fig. 13).

Reviewer 2

The authors present an interesting study modelling the transmission dynamics of avian influenza with parameters fit to empirical data from a live bird market. I find the model proposed to be reasonable; the statistical analysis appropriate; and the limitations of the approach to be unambiguously communicated. Generally, I find the manuscript to be uncommonly clearly written.

We thank the reviewer for their very positive assessment of our work.

Regarding the presentation of the model, I would encourage the authors to briefly state (perhaps any additional analysis is beyond the scope of this manuscript) how they would expect compartment topology to modify the parameter estimates or otherwise invest more time in the model description to motivate why SEEIRR is clearly the most appropriate framework.

We thank the reviewer for their suggestion. We now mention in the model description that our model is a variant of the SEIR model, which is commonly used to investigate AIV dynamics:

"This model is a variant of the more common SEIR model, which is typically used to investigate AIV dynamics (Fournié et Al., Mathematical Models of Infectious Diseases in Livestock: Concepts and Application to the Spread of Highly Pathogenic Avian Influenza Virus Strain Type H5N1, 2011)."

We added the new reference to the list of references. Furthermore, we commented on the choice of including two and one latent and infectious compartments, respectively:

"This is often regarded as a more realistic assumption than an exponential distribution of durations, which is implicit in models with single-staged compartments (Bouma et Al., PLOS Pathogens, 2009). On the other hand, including a single infectious compartment should not affect our results significantly since the main limitation to transmission within LBMs comes from the short length of stay.

[...]

The distinction between R^+ and R^- compartments allows to capture the persistence of viral RNA in infected chickens that recently stopped shedding (Griffin D.E., PLoS Pathogens, 2022)."

The reference to Griffin D.E., PLoS Pathogens, 2022 was also included in the list of references.

The authors describe several predicted epidemiological differences between exotic broilers and backyard chickens including:

“From our model’s output, we found a shorter latent period in exotic broiler compared to backyard chickens (Fig. 1B,C), lasting an average of 5.3 hours for exotic broiler, and 1 days for backyard chickens.”

and

“in the case of exotic broilers, most chickens with prior exposure to H9N2 were either infectious or latent, with only a minor proportion of them being immune (Fig. 1E). In contrast, most previously-exposed backyard chickens were immune to H9N2 (Fig. 1F). Our results thus tentatively suggest that prior infection occurs close to marketing age for broilers, whereas in backyard chickens it may occur further in the past, which is consistent with the latter being raised for a longer time compared to broilers.”

I encourage the authors to comment on whether they believe these findings are entirely consistent with variable patterns of prior exposure for these two populations or if the data presented suggest that there may be some unrecognized genetic determinants of susceptibility. Additionally I understand that to support the second passage quoted above, there has been an underlying assumption made that prior infection confers long lasting immunity (relative to the life cycle of a market chicken). If this is correct, I encourage the authors to explicitly state this assumption and perhaps introduce some additional background information regarding the market chicken life cycle so that the unfamiliar reader, like myself, may have some expectation of the maximum number of infections for one individual.

We thank the reviewer for this comment.

We have now provided some context regarding the ages at sale of broilers and backyard chickens in the Results section:

"These findings are consistent with known rearing practices and ages at sale of each chicken type: broilers are selectively bred to grow rapidly, and are sold for meat after just 28-31 days after hatching (Hennessey et Al., Preventive Veterinary Medicine, 2021). Backyard chickens are instead raised for meat and eggs in rural households and can reach much older ages. For context, backyard chickens in our dataset were aged between 90 and 720 days."

Our findings are also consistent with a previous study (Das Gupta et Al., *Transboundary And Emerging Diseases*, 2020) that found a higher seroprevalence of antibodies against H9 among backyard chickens in farms surrounding Chattogram, i.e. where the present LBM is located. While we cannot rule out the existence of unrecognised genetic determinants (which we already mention in the Discussion), our findings suggest that these are not necessary to explain observed differences in incidence in broilers and

backyard chickens. Nonetheless, the reviewer is right in pointing out that this interpretation rests on the role of pre-existing immunity. Ultimately, further confirmation about either hypothesis (which are not mutually exclusive) will require a better understanding of the role of previous immunity. We have discussed these aspects in multiple points in the Discussion:

"Our results indicate, however, that differences in earlier exposure to H9N2 AIV are sufficient to explain observed incidence patterns, and are consistent with known ages at sale and levels of H9 seropositivity in broilers and backyard chickens (Das Gupta et Al., 2021). A better understanding of the effectiveness of prior immunity, which we have assumed to be sterilising, will help validating or confuting this interpretation."

The authors consider both direct and environmental transmission, writing, "Ienv(t) accumulates due to shedding from infectious chickens and decays progressively at rate Θ . Here we consider three values of Θ , namely $\Theta^{-1} = 10, 3, 1$ days, corresponding to slow, intermediate and fast decay, respectively. These values are based on actual estimates from the scientific literature and capture a broad range of environmental conditions." I encourage the authors to briefly describe the physical nature of environmental transmission in a live bird market for readers, like myself, who are more familiar with influenza transmission dynamics in the human population which I expect substantially differ.

We thank the reviewer for their comment. We have now described the main sources of environmental contamination in the Results section:

"Within LBMs, the infection can be transmitted through poultry drinking water, as well as cages and floors that are contaminated by faecal material and during slaughtering. The lack of disinfection and the constant moving of cages and birds further promote poultry exposure to environmental contamination."

I find one of the more striking findings reported in the manuscript to be that while, "reduced length of stay and reduced probability of prior exposure, proved to be only modestly effective. Combining both interventions proved considerably more effective at reducing transmission compared to individual measures" I encourage the authors to clarify in the text if this effect is in fact nonadditive and if so to include some description of the expected mechanism behind the non-additivity.

The reviewer raises an interesting point that we address in the new Supplementary Fig. 14 shown below. Our reasoning is as follows: if the effect of each intervention in terms of reducing cumulative daily prevalence was additive, then their combined effectiveness E' would be given by $E' = E_1 + E_2$ (this of course holds up if and only if $E' \leq 1$).

Analogously, if their effects were multiplicative, the combined effectiveness would be $E' = 1 - (1 - E_1)(1 - E_2)$. To gauge any signatures of additivity and/or multiplicativity,

we measured the combined effectiveness from simulations E (panels A,B), and calculated the difference $E - E'$ under either expectation for E' (panels C-F). Interestingly, the positive effects of these interventions do not seem to combine additively or multiplicatively. In particular, the fact that $\Delta E > 0$ for most configurations, especially for stricter control of importations and shorter length of stay, is suggestive of a synergistic effect of stacking these interventions. This is mentioned in the Result section:

"Notably, the benefits of combining (i) and (ii) exceed expectations under the assumption that their effects were additive or multiplicative, hence suggesting a synergistic effect of multiple interventions (Supplementary Fig. 14)."

Supplementary Figure 14: Synergistic effects of combining interventions. This figure considers a simultaneous reduction of the maximum length of stay of chickens to T_m and the probability of prior exposure $\rho_{c,b}$ to $(1 - r)\rho_{c,b}$

(A,B). The effectiveness of these interventions combined is denoted with E , and is defined as the reduction in cumulative daily prevalence relative to a scenario with no intervention in place. The latter corresponds to $T_m = 5$ days and $r = 0$. The effectiveness of individual interventions is denoted as E_1 ($T_m < 5$ days and $r = 0$) and E_2 ($T_m = 5$ days and $0 < r \leq 1$). Panels C-F assess whether the effects of individual interventions add up in a multiplicative (C,D) or additive fashion (E,F). If the effect was purely multiplicative, the expected, combined effectiveness would be $\bar{E} = 1 - (1 - E_1)(1 - E_2)$. In the purely additive case it would be $\bar{E} = E_1 + E_2$. C-F show that $E > \bar{E}$ for most values of T_m and r , suggesting a synergistic effect of implementing these interventions simultaneously. Results are based on 2000 simulations from 200 independent samples from the posterior distribution. The first and second columns correspond to the fits to Ct=40 and Ct=33 data, respectively.

In conclusion, the authors present an interesting and well written study which I believe will be of broad interest.

Sincerely,

Nash Rochman (invited 12/5/23, returned 12/19/23)

Reviewer 3

The manuscript titled "Modelling the transmission dynamics of H9N2 avian influenza viruses in a live bird market" aims to understand the transmission dynamics of H9N2 avian influenza viruses in live bird markets (LBMs) in Bangladesh. The study uses a mechanistic transmission model, fitted to field data collected in a specific LBM, to simulate the spread of the virus among poultry.

The study presents significant findings on H9N2 transmission in LBMs, highlighting a rapid infection rate among susceptible chickens, often within a day. The virus's latent period is notably short, particularly in broiler chickens, sometimes as brief as 5.3 hours, facilitating its swift spread. It also uncovers considerable differences in exposure and immunity between exotic broiler and backyard chickens. Its unique approach, combining field data with mathematical modelling, and the robustness of its findings, underscored by sensitivity analyses and Bayesian methods, mark it as particularly relevant for areas where H9N2 is prevalent, guiding interventions to control its spread. Thus, This research offers crucial insights into the epidemiology of H9N2 in LBMs, vital for public health and the poultry industry.

I thoroughly enjoyed reading this manuscript and believe it is already of high quality. However, I suggest incorporating discussions on several interesting points. Firstly, the study's results are drawn from a single LBM in Bangladesh. It would be beneficial to explore how these findings might apply to other LBMs with varying dynamics, bird populations, and environments. Additionally, a more in-depth examination of environmental factors, like temperature, humidity, and market hygiene, is crucial as they significantly influence H9N2's survival and spread. A comparison with other avian influenza strains could also offer a broader perspective, highlighting differences and similarities in transmission dynamics. Finally, it would be interesting to consider whether

these findings could be applicable to livestock markets other than those for birds and discuss the generalizability of such high infection rates in group settings, dependent on the virus

We are pleased that the reviewer enjoyed our manuscript and thank them for their comments.

We have addressed the point on applicability to other LBMs in our reply to Reviewer 1.

We have now added a sentence to discuss potential applications to other LBMs, live animal markets and host-pathogen systems. There, we also argue for the need to account for the main AIV strain-specific characteristics and environmental drivers of transmission, such as the presence of suitable hosts and those factors that affect AIV survival in the environment (which are described in the Supplementary Methods).

"The ubiquity of similar poultry handling and trading practices suggests that our findings apply to other Bangladeshi LBMs as well. Applications of the model to other LBMs may require calibrating specific LBM features such as the number of chickens being traded and their length of stay. Applications to other AIV strains, e.g. H5N1, will also require accounting for the presence in the LBM of relevant hosts species (e.g. waterfowl in the case of H5N1 AIV), their specific infection parameters and differential ability to survive in the environment (Handel et Al., PLOS Comp. Biol., 2013). Finally, we note that our modelling framework could be applied to disentangle the contributions of external introductions and local transmission in other types of live animal markets and host-pathogen systems."

REVIEWERS' COMMENTS

Reviewer #1 (Remarks to the Author):

The authors have done an excellent job on their revisions and I have no further comments (and thank them for their respectful disagreement in regards to zoonotic potential - tbh, I think they are in a better position than I to judge this).

Reviewer #2 (Remarks to the Author):

I believe the authors have fully addressed the reviewer comments.

Sincerely,
Nash Rochman